# Adenoidal Disease and Chronic Rhinosinusitis in Children—Is There a Link?

**DOI:** 10.3390/jcm8101528

**Published:** 2019-09-23

**Authors:** Antonio Mario Bulfamante, Alberto Maria Saibene, Giovanni Felisati, Cecilia Rosso, Carlotta Pipolo

**Affiliations:** 1Otorhinolaryngology Unit, Department of Health Sciences, San Paolo Hospital, Università degli Studi di Milano, 20142 Milan, Italy; giovanni.felisati@unimi.it (G.F.); cecilia.rosso1@gmail.com (C.R.); carlotta.pipolo@googlemail.com (C.P.); 2Otorhinolaryngology Unit, San Paolo Hospital, 20142 Milan, Italy; alberto.saibene@gmail.com

**Keywords:** children, chronic rhinosinusitis, adenoid hypertrophy

## Abstract

Adenoid hypertrophy (AH) is an extremely common condition in the pediatric and adolescent populations that can lead to various medical conditions, including acute rhinosusitis, with a percentage of these progressing to chronic rhinosinusitis (CRS). The relationship between AH and pediatric CRS has been extensively studied over the past few years and clinical consensus on the treatment has now been reached, allowing this treatment to become the preferred clinical practice. The purpose of this study is to review existing literature and data on the relationship between AH and CRS and the options for treatment. A systematic literature review was performed using a search line for “(Adenoiditis or Adenoid Hypertrophy) and Sinusitis and (Pediatric or Children)”. At the end of the evaluation, 36 complete texts were analyzed, 17 of which were considered eligible for the final study, dating from 1997 to 2018. The total population of children assessed in the various studies was of 2371. The studies were categorized as surgical-observational, microbiological, genetic-immunological, and radiological. The analysis of the studies confirms the relationship between AH and CRS and supports the existing consensus on medical and surgical therapy. Furthermore, these studies underline the necessity to adapt medical and surgical treatment considering age, comorbidities including asthma and, if present, the Computed Tomography (CT) score.

## 1. Introduction

Adenoid hypertrophy (AH) is an extremely common condition in the pediatric and adolescent population, due to chronic inflammation that leads to the proliferation of adenoid lymphoid tissue [1], and is the primary cause of nasal obstruction in children [2]. According to a recent dedicated meta-analysis, AH prevalence is 34% in the general pediatric population and will vary between 42% and 70% when breaking down the general population into specific groups (children entering Ear Nose Throat (ENT) specialists, patients suffering from obstructive sleep apnea syndrome (OSAS), etc.) [3]. AH alone also seems to facilitate the appearance of other problems like oral respiration, OSA, speech alteration, otitis, and infection of the upper airways [4]. Viral infections of the upper airways and their possible evolutions are of particular interest due to their occurrence: 6% to 7% of upper respiratory viral infections may progress to adenoiditis and acute rhinosinusitis [5], with a percentage of these progressing to chronic rhinosinusitis (CRS) [6], condition characterized by 90 or more uninterrupted days of respiratory symptoms, including cough, nasal discharge, or nasal obstruction [5]. A recently published study involving 42.1 million school-age pediatric patients found a 4% annual incidence of CRS, corresponding to an affected 1.7 million school-age children [7], with a prevalence in the general population from 2.7% to 6.6% [8] The exact prevalence of CRS in children is difficult to determine because only a small percentage of cases are reported to the physician [9]. CRS also has an important effect on the quality of life [10], both directly and indirectly; several studies consider CRS to be an important risk or precipitating factor of chronic respiratory diseases, above all asthma [7,9,11]. Although active debate persists concerning the best medical and surgical therapeutic strategies to be applied [12], the clinical consensus statement on the optimal diagnosis and management of pediatric chronic rhinosinusitis of 2014 [6] and the international guidelines [13,14,15] suggest as a first step to treat the patient with an empirical antibiotic therapy effective against typical rhinosinusitis pathogens for at least 20 consecutive days. If the condition persists, culture-directed antibiotic therapy is to be prescribed. Furthermore, good clinical practice is to implement a therapy with a topical nasal saline wash and a topical nasal steroid. If antibiotic treatment is not successful, adenoidectomy is considered the first line surgical procedure for children up to 12 years old. Endoscopic sinus surgery (ESS) should be considered only in case of failure of medical therapy and adenoidectomy. 

This is the therapeutic approach justified by the relationship existing between AH and pediatric CRS and the objective of this study is to perform a systematic review of the literature and data on this relationship.

## 2. Experimental Section

A systematic literature review was carried out in June 2019 through Web of Science, PubMed, and Scopus databases, using a search line for “(Adenoiditis OR Adenoid Hypertrophy) AND Sinusitis AND (Pediatric OR Children)”. After completion of all searches, duplicates were eliminated. In vivo and in vitro studies were included based on the causal relationship between adenoiditis or AH and pediatric CRS, regardless of whether they were prospective or retrospective. Both anatomical, observational, therapeutic, and microbiological studies were considered eligible for the study. Two of the authors (AB and CP) independently screened the retrieved studies based on the title, keywords, and abstract to exclude non-relevant and non-English written studies. Both retrospective and prospective studies were included, while small series and case reports were excluded due to their intrinsic lower level of evidence (the minimum number of patients was arbitrarily set to 10). Published reviews and textbooks on the subject were excluded, but their reference list was reviewed to identify possible additional studies. Studies whose main purpose was the relationship between AH and acute rhinosinusitis were excluded, as were studies where AH and CRS were considered but not compared (for example the impact of AH and CRS separately on asthma). Studies that did not meet the inclusion criteria were discarded during the initial review. When uncertainty existed in the abstract evaluation, the full text of the document was retrieved and assessed. A manual search in the reference lists of these articles was performed to identify potentially relevant papers missed during the database search. Differing opinions were resolved by consensus between the two authors. Data extracted and analyzed for the study included mainly the statistically-significant relationship between AH and CRS.

## 3. Results

At the end of the evaluation, 36 complete texts were analyzed, 17 of which were considered eligible for the final study; the 17 texts contained the results of 18 studies. The search results are shown in the PRISMA chart (Figure 1). The 17 studies were analyzed in chronological order, from the oldest (1997) to the most recent (2018). The total population assessed in the various studies was of 2371 children (range 16–817) with an average age of 6.6 years. The Male-Female ratio (M:F) was extremely variable between studies, but in none of them a significant gender predisposition was identified. The authors divided the 18 studies into categories according to the method used to assess the relationship between AH and CRS (see Table 1): 6 surgical–observational studies, 8 microbiological studies, 3 genetic-immunological studies, and 1 radiological study. The surgical-observational category contains studies that are based on evaluation of CRS improvement after therapy, both anti-biotical and surgical. The microbiological studies focus on the analysis of bacterial strains isolated from adenoids, nasal or sinus swabs, and samples. Some of these studies made use not only of cultures of bacteria but also tested several antibiotics. The genetic-immunological studies investigate different alterations of inflammatory and innate immunity pathways. The radiological study investigates the role of a CT score in predicting the risk of failure of adenoidectomy in CRS children.

Five out of 6 observational surgical studies demonstrate a statistically significant relationship between adenoid pathology and CRS, the only one that does not demonstrate a link (Wang D. et al. – 1997) investigates the relationship between simple adenoid size and CRS.

Seven out of 8 studies focusing on the assessment of common colonization/bacterial infection of adenoids and paranasal sinuses demonstrate a significant presence of the same bacterial strains on both adenoids and paranasal sinuses, underlining the significant relationship between adenoid disease and CRS. One study (Cedeño E. et al. -2016) had the objective of identifying *H. pylori* and did not demonstrate the colonization of *H. pylori* strains in either adenoids or in sinuses. 

All three of the genetic/immunological studies have shown alterations in the expression of different genes and factors of the inflammatory pathway in children with adenoiditis and CRS; the expression of all factors is instead normal in the absence of CRS. 

The radiologic study shows the frequent coexistence of AH and CRS and the greater therapeutic challenge compared to a simple CRS.

## 4. Discussion

AH is a widespread condition in the pediatric age [3,4] The possible evolution towards nasal or sinus infections and the relationship among these conditions is well-known and studied [16]. The value of establishing a relationship between AH and CRS is related to the possible therapeutic consequences. The first line therapy of AH and CRS is medical, based on high volume nasal lavage with a saline solution, corticosteroid nasal spray, and systemic antibiotic treatment in case of acute bacterial infection; in case of failure after maximal therapy, surgery is necessary [3,6,17]. International guidelines advocate for a first line isolated adenoidectomy, followed by endoscopic sinus surgery only in case of treatment failure [13,14,15,16]. The indication to perform at first a simple adenoidectomy (tonsillectomy is not indicated) is due to the strong relationship that exists between AH and CRS. Some authors prefer to speak of relationship between CRS and adenoiditis [18,19,20] rather than between AH and CRS, but most prefer the latter, in consideration of the etiology of AH as a likely consequence of chronic adenoiditis [17,21,22,23]. Over the past few years, several authors demonstrated the correlation between AH and CRS through different types of studies, based on the endoscopic study of the adenoid volume [24] on the same bacterial strains both on the adenoidal tissue and in the nasal sinuses [20,22,23,25,26,27,28] on the positive effects on CRS by adenoidectomy [21,29,30,31] on specific immunologic modifications [32,33,34] and on radiological findings [35]. The most important data that derives from the many studies is the multifactorial aspect of the relationship between adenoid and sinus pathology. 

### 4.1. Surgery

As explained previously, adenoidectomy is considered the first surgical treatment to be performed for pediatric CRS. This indication derives from several studies focused on the beneficial effect of adenoidectomy both for obstruction and bacterial reservoirs removal. One of the first studies has been published in 1997 by Wang and colleagues [24] and assessed whether the simple dimension of the adenoids were a risk factor for RS, enlisting a cohort of 817 children of which 89 were affected by purulent rhinosinusitis. However, only 15 out of 89 (16.9%) were affected by AH, showing no relationship between sinusitis and simple adenoid size. This data was particularly meaningful in children between 1 and 2 years old and between 3 and 4 years old. We can assume this insufficient correlation is a consequence of the inclusion criteria of the study, as the diagnosis of purulent sinusitis was already done when pus was detected under one of the turbinates and/or sinus involvement was confirmed by CT scan, enrolling both acute and chronic rhinosinusitis. AH and acute rhinosinusitis may not be as strictly correlated as AH and CRS.

While Wang’s group focused only on simple AH and the consequent nasal stenosis, in 2003 Ungkanot and colleagues [29] investigated the importance of adenoidectomy to eradicate the reservoir of infection in the vicinity of the sinus ostia. They selected a population affected by recurrent rhinosinusitis, and they evaluated the mean episodes of disease before and after surgery as a function of the number of medical examinations needed due to the occurrence of new episode of symptoms. The authors observed a significant reduction in episodes of disease, from 13.7 before to 0.76 per year after adenoid surgery. From this data, the conclusion of the study was that adenoid surgery is to be carried out prior to endoscopic sinus surgery, as later also indicated in the clinical consensus statement on the optimal diagnosis and management of pediatric chronic rhinosinusitis of 2014 [6]. The role of adenoidectomy as an instrument to remove bacterial reservoir is supported also by the study by Tosca et al [19] that correlates parameters including nasal cytology and microbiological cultures with nasal endoscopy findings and post adenoidectomy outcome; they demonstrated once again that the correlation between chronic rhinosinusitis, adenoiditis, and microbiology is significant. 

In 2007 Ramadan and colleagues analyzed 143 children that had adenoidectomy for CRS and observed that around 50% required a subsequent ESS because of symptom persistence at an average of 24 months after surgery, especially in children who had asthma and were of a younger age (7 years old or less) [30] This may be due to the normal adenoids involution during growth and to the role of the inflammation pathway activated in an asthmatic patient, but the high demand for a second intervention guided the same group to perform another study in 2008, where two cohorts of children with chronic sinusitis differentiated by severity of their CT score were considered, assessing that sinus washes added to adenoidectomy would have improved the likelihood of a successful surgical outcome [21] They observed a success rate of 87.5% for the group that included sinus lavage and 60.7% for the group performing only adenoidectomy during the 12 months following surgery. Children with a higher CT score (greater or equal to 6) had a better outcome compared to those undergoing only adenoidectomy. The children who had a low CT score (less than 6) did not have a statistically better outcome if a sinus wash was added at the time of adenoidectomy. Despite these results, sinus wash nowadays is not considered a golden standard procedure and in the Clinical Consensus Statement the authors did not reach consensus on antral irrigation as either a sole therapy or as an addendum to adenoidectomy. No consensus either has been reached on the implementation of alternative sinus procedures other than endoscopic sinus surgery, as for example balloon sinusoplasty. Nevertheless, this last technique has been studied by Gerber and colleagues [31] as a possible integration to adenoidectomy in children with CRS. The success rate of the procedure was compared to washing of the maxillary sinuses by puncture. During the follow-up, they investigated changes in the quality of life of children; the study showed no significant difference in success rate between the two procedures, concluding that using a more expensive procedure, albeit less invasive, was not indicated routinely. Nevertheless, it can be considered a safe therapeutic option in specific cases.

### 4.2. Microbiology

The studies which will be discussed below had the objective of understanding if the bacteria causing chronic adenoid infection was the same as that causing CRS. The study of the group of Bernstein and colleagues [25] is of particular interest, as it has shown that in 89% of cases with CRS it was possible to isolate the same bacterial colonies from swab collection on adenoids as on the side wall of the nose. *Haemophilus influenzae*, *Streptococcus pneumoniae*, and *Moraxella catarrhalis* were the most frequent bacterial strains isolated, while *Staphylococcus aureus* was excluded in consideration of the high rate of nasal colonization. In 2008 Shin and colleagues [27] performed a similar investigation, making use of adenoid samples instead of swabs. They isolated similar bacterial strains as Bernstein and colleague (*Haemophilus influenza*, *Streptococcus pneumonia*) but also *Streptococcus pyogenes* and *Staphylococcus aureus*. They also analyzed preoperative paranasal X-ray and showed the bacterial isolation rate increased significantly with sinusitis grade, concluding that there is indeed a correlation between CRS severity and adenoids infection and that *Staphylococcus aureus* is not only a colonizing bacterium. Completely focused on this latter bacterium is the study by Lin and colleagues [20] that analyzed a cohort of 283 children in Taiwan, showing the presence of *Staphylococcus aureus* in 21.2% of specimens from chronic adenoiditis, 35% of which were MRSA. They also demonstrated that *Staphylococcus aureus* was frequently a cause of AH and not just simply a colonization. All these studies helped define the bacterial strains that are commonly involved in adenoiditis, AH and consequently in pediatric CRS, that are *Haemophilus influenzae*, *Streptococcus pneumoniae*, *Moraxella catarrhalis*, *Streptococcus pyogenes*, and *Staphylococcus aureus*. This information is helpful to establish an empiric first line therapy to treat pediatric CRS. 

Further consolidation of this therapeutic approach was provided by Davcheva-Chakar and colleagues [28] who tested the susceptibility of microorganisms isolated from adenoid and sinus samples to the penicillin group, cefadroxil, cefpodoxime, ceftriaxone, cefotaxime, aminoglycoside, clindamycin, macrolides, quinolone, and cotrimoxazole. They showed that *Haemophilus influenzae*, *Streptococcus pyogenes*, and *Staphylococcus aureus* strains were susceptible to all these antibiotics, except for cotrimoxazole. Mild susceptibility and resistance to certain antibiotics was found for *Streptococcus pneumoniae*, and *Moraxella catarrhalis* isolates. 

Several research groups over the last 10 years focused on the identification of other bacterial strains that could shed new light on CRS etiology and on the relationship between AH and CRS. An interesting study by Nia and colleagues focused on *Clamydophila pneumoniae*, this uncommon infection can lead to CRS in the pediatric population, particularly in middle east countries [22]. The group demonstrated that adenoids can act as a reservoir for *Clamydophila pneumoniae*, and cause rhinosinusitis concomitant with AH. They suggested performing target antibiotic therapy before adenoidectomy, to increase the surgical success rate. The authors underlined that this type of infection is rare in other parts of the world, but the data is still of great interest considering modern world complex migration. A further study carried out by the Cedeño group [23] focused instead on the search for *Helicobacter pylori* at the nasal, sinus, and adenoid level, driven by the assumption that gastroesophageal reflux is a risk factor for AH and CRS, even if there is no consensus in literature regarding this relationship [6]. Unfortunately, the authors were never able to identify the organism.

In recent years the attention of researchers has shifted from individual bacterial strains to biofilms, complex habitats where different bacteria organize and coexist within polysaccharide matrices that provide them with a solid defense against antibiotics and other external aggressions [20,26]. In 2007, Coticchia and colleagues performed a comparative microanatomical investigation of adenoid mucosa from children with CRS and obstructive sleep apnea syndrome (OSAS); they studied biofilm distribution using scanning electron microscopy. One of the most interesting results they obtained was that adenoid samples from children with CRS had a dense mature biofilm that covered almost 95% of the mucosa surface, while only 1.9% of the mucosal surface of children with OSA was covered by biofilm [26]. Given the solid organization of the biofilm, the same authors considered adenoid surgery to be useful as a tool not only for volume reduction but also for the elimination of bacterial reservoirs.

Considering the previous data, it is possible to conclude that the most frequent bacterial strains are *Haemophilus influenzae*, *Streptococcus pneumoniae*, *Moraxella catarrhalis*, *Streptococcus pyogenes*, usually susceptible to common antibiotic, and *Staphylococcus aureus*, unfortunately often MRSA. Therefore, good clinical practice should include starting with a wide range empiric antibiotic therapy [6] and in case of failure to prescribe a culture-directed antibiotic therapy, looking for antibiotic resistance or atypical bacterial strains. 

### 4.3. Radiology

The effectiveness of adenoidectomy ranges between 47% and 56%. It has been shown to be less effective in children who have asthma with CRS. Why adenoidectomy is more effective for certain children and less for others appears complex and at times mystifying [35,36,37]. It is reported in literature that children symptoms of CRS and AH are similar and include nasal stuffiness, nasal discharge, cough, and headache in older children. In 2004, Bhattacharyya et al. [38] noted a distinction can be made on the Lund-Mackay computed tomography (CT) score. They suggested that a CT score of 5 is indicative of CRS. In 2014 Ramadan and colleagues investigated the power of this CT score as an outcome predictor for adenoidectomy in children affected by CRS: a diagnosis of CRS was given to those children with symptoms of CRS not responsive to medical treatment and having a CT score over 4; a diagnosis of AH was given for the remaining children. They demonstrated that adenoidectomy was helpful for children with AH, particularly if non-asthmatic patients, while children with CRS usually had a more severe outcome. The authors concluded that making the diagnosis of CRS in children with CT scan is critical in determining whether, initially, an adenoidectomy alone is an appropriate treatment specifically for those who have asthma. In literature, use of a CT scan in children affected by CRS is accepted, but usually is indicated only prior to ESS, to assess structure, development and extent of the disease, particularly in recurrent cases and in patients with extensive nasal polyposis that can distort anatomical landmarks. It should be noted that ESS is indicated only if medical therapy, adenoidectomy, or both have failed, or in other specific cases. Using CT as the first line diagnostic instrument and in adenoidectomy planning is not recommended, because it provides limited information on adenoid size alone, which does not necessarily correlate with ability to serve as a bacterial reservoir for infection. Moreover, imaging studies involve radiation of the skull and brain, which carries a postulated risk of malignancy. As a first line diagnostic tool, nasal endoscopy is to be preferred. 

### 4.4. Genetic/Immunology

Some groups have focused on the analysis of immunological changes and inflammatory responses in children that have AH +/- CRS [32,33,34] demonstrating alterations in various pathways (mainly membrane proteins, IgA and tissue-remodeling–associated cytokines). These studies are preliminary but provide interesting results that suggest the possibility to develop in the future targeted biological drugs.

A first study by Eun and colleagues [34] evaluated IgA, IgG, IgD, and IgM concentrations in three groups (AH, CRS, and Otitis Media with Effusion OME) using the immunohistochemical technique. They also compared the levels of expression of the B lymphocyte inducer of maturation program 1 (BLIMP-1), a promoter of plasmacytosis, and B cell leukaemia/lymphoma-6 (BCL-6), a repressor of plasmacytosis, in the adenoids of these children. The group showed a reduction in expression of Ig A and of antibody to BLIMP-1 in the CRS and AH groups. Staining scores of antibodies to BCL-6 did not differ significantly among the three groups. Reductions in IgA in serum may increase the susceptibility to recurrent infection, including CRS, and it is probably not due to increased clearance caused by inflammatory reactions. Rather, the increase susceptibility to infection is caused directly by the reduction of IgA. They also found BLIMP-1 expression in adenoid tissues was lower in CRS than in AH groups. The reduced expression of BLIMP-1 in CRS groups was associated with the reduction in IgA secreting cells, thus decreasing immunity to viruses, bacteria, and other antigens, and increasing the likelihood of CRS.

In 2015, Qu and colleagues [32] investigated another aspect of the immunity alteration that correlates AH with CRS; they demonstrated that adenoid samples from CRS children had a lower level of surfactant proteins A (SP-A ) and D (SP-D); these are hydrophilic proteins belonging to the Collectin family of innate immunity proteins and are secreted from epithelial surfaces; they bind to carbohydrate moieties present on the surface of bacteria, fungi and viruses and lead to the clearance of pathogens by antigen-presenting cells (e.g., macrophages and dendritic cells). If adenoidal epithelial cells in CRS patients secrete less SP-D than in AH patients, their innate immune defense would be compromised. Therefore, CRS patients are inflicted with persistent infections and subsequent inflammation. The group proposed that for this reason, AH is often comorbid with CRS. While this group focused on decreased immunity, the group of Shin and colleagues [33] investigated the role of the inflammation pathway. They analyzed adenoid tissue homogenates from 16 children with CRS and from 24 children without CRS to quantify the levels of inflammatory cell activation markers, including soluble interleukin (IL)-2 receptor (sIL-2R), soluble CD23 (sCD23), IL-6, eosinophilic cationic protein (ECP), and tryptase, and the levels of cytokines associated with tissue remodeling, like transforming growth factor (TGF)-1, matrix metalloproteinase (MMP) 2 and 9, and tissue inhibitor of metalloproteinase (TIMP)-1. The mean levels (the ratio to albumin level) of sIL-2R, TGF-_1, MMP-2, MMP-9, and TIMP-1 were significantly higher in adenoid tissues of patients with CRS. Regarding the severity of CRS, the ECP level was significantly higher in patients with severe CRS than in those with mild to moderate CRS. Adenoid tissues in pediatric CRS patients had higher levels of tissue-remodeling–associated cytokines, which may explain the relationship between pediatric CRS and adenoid inflammation. 

As described before, all these papers are to be considered preliminary studies and more research is necessary to improve the knowledge of this complex inflammatory pathway and the relation between their activation, CRS and AH. However, this data underlines when CRS is present, the adenoid samples show a decrease of immunity factors like IgA or antigen presenting protein cells that is probably the causing agent of bacterial strains proliferation and consequently of CRS. Furthermore, the proliferation of bacteria leads to the activation of an inflammation pathway that causes the adenoids to increase in volume (AH). All these conditions lead to the creation of a loop in which the occurrence of infections and inflammations increases and is associated to volume augmentation of the adenoids, defining the clinical presentation of a typical pediatric CRS patient. 

## 5. Conclusions

Analysis of the above studies confirms the relationship between adenoid hypertrophy and CRS and confirms also that the medical and surgical therapy aimed at the treatment of the former is effective on the treatment of the latter. However, albeit not represented in the Clinical Consensus Statement^6^ and in clinical guidelines [13,14,15], these studies further underline the necessity to personalize medical and surgical treatment, taking into consideration age, comorbidities including asthma and, if present, the CT score.

## Figures and Tables

**Figure 1 jcm-08-01528-f001:**
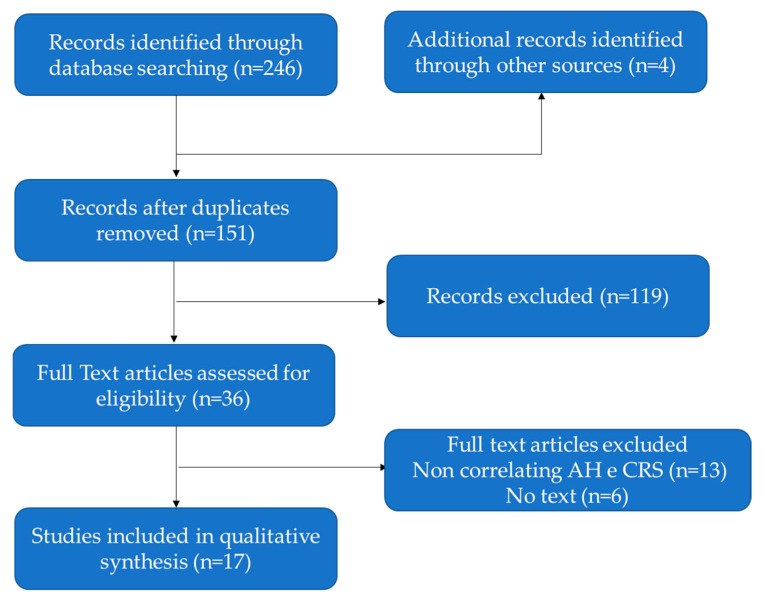
PRISMA chart showing the selection process of the studies included in this study.

**Table 1 jcm-08-01528-t001:** Studies reporting results AH-CRS relationship.

Author, Year	Study Design	Patients	Main Topic	AH - CRS Relationship	Summary
Wang D. et al. (1997)	Retrospective	817	S/O	NS	Evaluation of simple dimension of the adenoids as risk factor for CRS; only 16.9% of patients presented AH, showing negative relation between CRS and simple adenoid size.
Bernstein J. et al. (2001)	Prospective	52	M	S	In 89% of cases with CRS isolation of the same bacterial strains from adenoids and lateral nasal wall: *H. influenzae, S. pneumoniae* and *M. Catarrhalis*.
Tosca M. et al. (2001)	Prospective	145	S/O + M	S	The correlation between chronic rhinosinusitis, adenoiditis, and microbiology is very significative, based on nasal cytology, microbiological cultures, nasal endoscopy and surgical outcomes.
Ungkanont K. et al. (2004)	Prospective	37	S/O	S	Importance of adenoidectomy to eradicate the reservoir of infection in the vicinity of the sinus ostia. The conclusion of the study was to consider adenoid surgery prior to ESS.
Coticchia J. et al. (2007)	Prospective	16	M	S	Adenoid biofilm distribution in CRS vs. OSAS. In CRS samples, a dense uniform biofilm covering almost all adenoidal tissue was present, while in the OSAS group only scattered area of biofilm.
Ramadan H. et al. (2007)	Retrospective	55	S/O	S	The authors demonstrated that > 50% of children with CRS that underwent adenoidectomy would require an ESS because of symptoms persistence at an average of 24 months after adenoidectomy; asthma and <7 yo are risk factor.
Ramadan H. et al. (2008)	Retrospective	60	S/O	S	Antral wash during adenoidectomy improves surgical success rate. They showed a higher success rate for wash/A group than for the adenoidectomy group after at least 12 months after surgery.
Shin K. et al. (2008)	Retrospective	410	M	S	Bacteria were isolated in 79.3% of cases: 28.5% *H. influenzae*, 21.7% *S. pneumoniae*, 21.0% *S. pyogenes*, 15.6% S. *Aureus*, 4.4% *MRSA*, and 7.6% other species.
Eun Y et al. (2009)	Prospective	79	G/I	S	Authors evaluated IgA, IgG, IgD, IgM, BLIMP-1, and BCL-6: reduction in expression of Ig A and of antibody to BLIMP-1 in the CRS and AH groups; probably the susceptibility to infection is caused by the reduction of IgA.
Shin S. et al. (2009)	Prospective	40	G/I	S	Levels of inflammatory cell activation markers were significantly higher in adenoid tissues of patients with CRS. Levels were significantly higher in patients with severe CRS than in those with mild to moderate CRS.
Lin C et al. ( 2012)	Prospective	283	M	S	Cohort of children of Taiwan: *S. Aureus* was present in the 21.2% of specimens from chronic adenoiditis and 35% was *MRSA*. *S. Aureus* is frequently a cause of AH and not only a simple colonizer.
Ramadan H. et al. (2014)	Retrospective	233	R	S	CT as an outcome predictor for adenoidectomy in children affected by CRS. Adenoidectomy was very helpful for children with AH, while children with CRS usually had the worst outcome.
Nia S. et al. (2014)	case control	53	M	S	*Clamydophila pneumoniae* can lead to CRS, particularly in middle eastern countries. Adenoids can act as reservoir for C. Pneumoniae and cause CRS concomitant with AH.
Davcheva-Chakar M. et al. (2015)	Prospective	20	M	S	*H. Influenzae, S. Pyogenes, S. Aureus, S. Pneumoniae,* and *M. Catarrhalis* susceptibility to antibiotics. Mild susceptibility and resistance to antibiotics have been found for *S. Pneumoniae* and *M. catarrhalis*.
Qu X. et al. (2015)	case control	18	G/I	S	Adenoid samples from CRS children had lower lever of surfactant protein A (SP A) and D (SP D), these are hydrophilic proteins of innate immunity; they lead to the clearance of pathogens by antigen presenting cells.
Cedeño E. et al. (2016)	Prospective	28	M	NS	*H. pylori* at nasal, sinus, and adenoid level, considering gastroesophageal reflux as a risk factor for AH. Authors were almost never able to identify the organism.
Gerber M. et al. (2018)	Prospective	25	S/O	S	Balloon sinusoplaty as a possible integration to adenoidectomy in children with CRS; The procedure was compared with the washing of the maxillary sinuses by puncture. The study showed showed no significant differences between the two procedures.

S = Significant, NS = Non-Significant, S/O= Surgical/Observational, M = Microbiological, G/I = Genetic/Immunologic, R = Radiological; CRS, chronic rhinosinusitis; OSAS, obstructive sleep apnea syndrome; AH, Adenoid hypertrophy; ESS, Endoscopic sinus surgery.

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
