# Peer review of "Adenoidal Disease and Chronic Rhinosinusitis in Children—Is There a Link?"

_jcm, 2019, doi:10.3390/jcm8101528_

Round 1
Reviewer 1 Report
A contemporary review of adenoidal hypertrophy and CRS - a common clinical problem. This describes clinical and scientific research of the topic and highlights the current knowledge gaps.
Author Response
I would like to thank the reviewer for his time and positive opinion on our work.
I have proceeded to subdue the manuscript to a native English speaker that has extensively modified the text.
I hope that this new version will be clearer and easier to be read.
Kind regards
Reviewer 2 Report
The authors present a systematic literature review assessing the association between adenoid hypertrophy and chronic rhinosinusitis.
Recommend that the authors proofread the manuscript prior to submission as this has detracted greatly from being able to review the paper adequately. Significant changes to the text are required – usage of commas, avoidance of colloquialisms.
The study methodology is sound, and the relevant literature is covered. Certainly the data/review could be better synthesised but overall ok
Major revision is required before acceptance.
Certain examples provided below
Shorter paragraphs
Line 207: it’s not a simple colonizing bacterium
Line 211: avoid colloquialisms such as “even” - They even demonstrated that Staphylococcus Aureus was frequently a cause of AH and not only responsible for a simple colonization
Line 224 does not make sense
Line 261: capitalisation
Inappropriate capitalisation of words and some not – see line 200-201 for bacterial names; more standardisation across the text required.
Author Response
Dear Editor and Reviewer,
We are submitting the reviewed version of our article “Adenoidal disease and chronic rhinosinusitis in children. Is there a link?”.
Thank you for your interesting comments and for your efforts to improve our manuscript.
Recommend that the authors proofread the manuscript prior to submission as this has detracted greatly from being able to review the paper adequately. Significant changes to the text are required – usage of commas, avoidance of colloquialisms.
Line 207: it’s not a simple colonizing bacterium
Line 211: avoid colloquialisms such as “even” - They even demonstrated that Staphylococcus Aureus was frequently a cause of AH and not only responsible for a simple colonization
Line 224 does not make sense
Following the reviewer indications, we have we submitted the manuscript to a native English speaker that has extensively modified the text, removing colloquialism and inappropriate language. The main modifications are underlined in the text in order to make easier the review.
Inappropriate capitalisation of words and some not – see line 200-201 for bacterial names; more standardisation across the text required.
We have proceeded to a complete review of the manuscript to improve standardisation, particularly concerning bacteria nomenclature.
We hope that you will appreciate the manuscript's improvement and that you’ll find it worthy of your readership. Thank you again for your insightful suggestions.
Kind regards